# Neural tuning instantiates prior expectations in the human visual system

William J. Harrison [1,2], Paul M. Bays[3] & Reuben Rideaux [2,3,4] ✉

Perception is often modelled as a process of active inference, whereby prior expectations are combined with noisy sensory measurements to estimate the structure of the world. This mathematical framework has proven critical to understanding perception, cognition, motor control, and social interaction. While theoretical work has shown how priors can be computed from environmental statistics, their neural instantiation could be realised through multiple competing encoding schemes. Using a data-driven approach, here we extract the brain's representation of visual orientation and compare this with simulations from different sensory coding schemes. We found that the tuning of the human visual system is highly conditional on stimulus-specific variations in a way that is not predicted by previous proposals. We further show that the adopted encoding scheme effectively embeds an environmental prior for natural image statistics within the sensory measurement, providing the functional architecture necessary for optimal inference in the earliest stages of cortical processing.

The human visual system is tasked with inferring environmental attributes from image data that can be corrupted by noise from both internal and external sources. To combat the influence of such noise, a statistically optimal, or ideal, observer would compute the distribution of possible environmental states, and combine this prior information with incoming sensory signals to infer the true state of a scene[1]. While this general framework for understanding sensory encoding is relatively uncontroversial, the biological instantiation of the prior for environmental statistics is largely mysterious.

Sensory representations are thought to be tuned to behaviourally relevant statistics of natural environments over evolutionary and developmental timescales[1–4]. As shown in Fig. 1a, edges and contours in natural images are primarily oriented along the cardinal axes[5–8]. There is a corresponding anisotropy in the orientation selectivity of visual neurons in several mammalian species that prioritises the encoding of cardinally oriented information (Fig. 1b)[9–11]. Analogously, humans are superior on a range of visual tasks for stimuli that are oriented around cardinal orientations relative to oblique orientations[12]. Such biases in encoding and behaviour are even present in artificial intelligence systems trained on naturalistic movies[13,14]. Several computational accounts have attempted to unify the influence of environmental

statistics on the properties of sensory neurons as well as perception[5,15,16], but have been unable to address empirically how such encoding is implemented at the neural level.

One way in which environmental regularities could be represented by biological systems is via sensory encoding schemes that allocate neural resources in relation to the natural frequency of various features. However, such models are consistent with multiple competing encoding schemes. In the domain of visual orientation, for example, biases in perception can be explained by corresponding biases in either the width *or* spacing of sensory tuning curves[5,15–18]. Therefore, while it is generally accepted that perceptual biases can be linked to anisotropies in environmental statistics, there is far less clarity about how these anisotropies are represented in the human visual system (as discussed in[19]). Moreover, leading models that account for cardinal biases in perception equate the prior probability of horizontal and vertical features[5,15,16], whereas earlier work suggests that horizontal features are over-represented relative to vertical[6,20]. These discrepancies may obfuscate a clear understanding of neural encoding schemes; we are not aware of any prior work that has attempted to link such biased environmental statistics with population-level responses of the human visual system. The way in which environmental priors are

[1]School of Psychology, The University of Queensland, St Lucia, Australia. [2]Queensland Brain Institute, The University of Queensland, St Lucia, Australia. [3]Department of Psychology, The University of Cambridge, Cambridge, UK. [4]School of Psychology, The University of Sydney, Camperdown, Australia. ✉e-mail: reuben.rideaux@sydney.edu.au

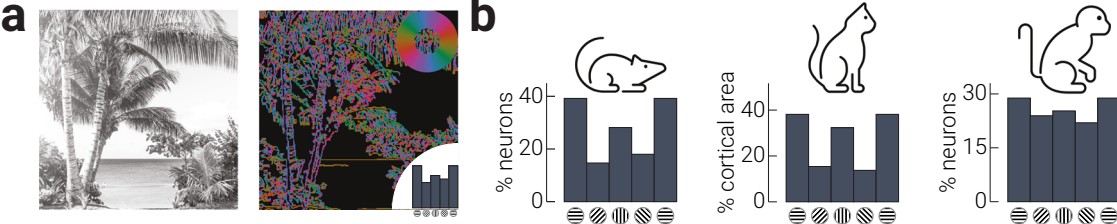

**Fig. 1 | Neural tuning anisotropies mirror natural image statistics. a** Example image of natural scene (left) and its composite orientations (right). Orientations are coloured according to inset colour wheel; summary of relative orientation distribution indicated by inset histogram. **b** Distribution of orientation selectivity in primary visual cortex of mouse[10], cat[11], and macaque[9]. Beach photo credit: Matthew Brodeur, https://unsplash.com/photos/DH_u2aV3nGM.

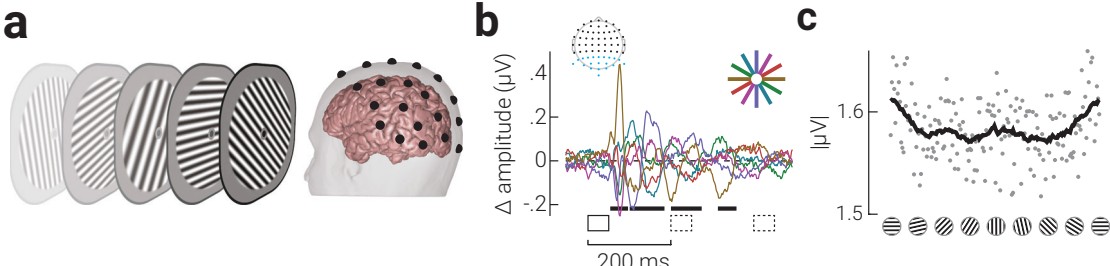

**Fig. 2 | Anisotropic univariate neural responses evoked by oriented gratings. a** Schematic showing the experimental design; observers viewed rapidly presented oriented gratings (neural probe) while monitoring for target gratings with lower spatial frequency. Following the neural probe, participants indicated the number of target gratings detected using the mouse. **b** Difference in event related potentials for gratings within orientation bins ([0°, 30°, 60°, 90°, 120°, 150°] ±15°) from the grand average of responses, averaged over responses to all gratings. Signals averaged across parietal and occipital EEG sensors (blue dots on topographic map) and participants (n = 36). Orientation bin indicated in colour; event-locked and subsequent gratings indicated by solid and dashed black rectangles, respectively, and cluster-corrected periods of significant difference between orientations indicated by horizontal black bars. **c** Time-averaged absolute signal amplitude for each grating orientation, binned to the nearest whole degree. The black line indicates the moving averages of data points.

represented in biological systems is a critical gap in our understanding of how the brain achieves optimal inference.

In the present study, we investigated the instantiation of environmental priors in the human visual system using electroencephalography (EEG) and inverted encoding models[21,22]. We developed novel neural decoding and generative modelling procedures with which to infer anisotropic coding in the human visual system, and compared these anisotropies with responses generated from encoding schemes with known neural response functions. Our approach therefore provides a generalisable means to test arbitrarily specified sensory encoding schemes. We show, however, that no previously proposed encoding scheme can explain the empirical neural data. Instead, we show that anisotropic neural responses can be explained almost entirely by a redistribution of sensory tuning curves that prioritises the most dominant environmental structure. We further show that the recovered neural code embeds a prior for natural image statistics within the sensory measurement, thereby simplifying the biological instantiation of optimal inference. Our results thus support the biological plausibility of perception as Bayesian inference by explaining how prior expectations are encoded by the tuning properties of sensory neurons.

## Results
### Decoded neural responses over-represent horizontal
We recorded human observers' brain activity with EEG while they viewed rapidly presented oriented gratings (randomly sampled from a uniform distribution between 0–180°) and monitored for occasional changes in spatial frequency (Fig. 2a). Prior to the main analyses of the neural activity evoked by the gratings (Fig. S1), we established that orientation information was primarily represented in parietal and occipital EEG sensors (Fig. S2; *Method - Neural Decoding*); thus, we only included the signal from these 20 sensors in all subsequent analyses

(inset of Fig. 2b, cyan dots). We first characterized orientation-related univariate activity. We sorted the gratings into six orientation bins (±15° around 0°, 30°, 60°, 90°, 120°, and 150°, where 0° is horizontal) and calculated the difference between the average evoked response for each bin from the grand average of responses, averaged over responses to all gratings. As shown in Fig. 2b, there were significant differences in the response to orientations between approximately 50 to 400 ms following stimulus onset. The largest deviation from the mean response to all gratings was for horizontal orientations, which occurred in the initial stages of stimulus processing (50–100 ms following stimulus onset). We then collapsed the responses across time and calculated the average absolute response for each grating orientation and found that response amplitude peaked around horizontal gratings (Fig. 2c).

We next quantified temporal dynamics in orientation tuning using inverted encoding analyses[21]. Using cross-fold validation, we trained and tested an inverted model using the EEG signals to decode the orientation of each grating from the neural activity at each time point. From the trial-by-trial decoded signals, we derived summary parameter estimates of *accuracy* (the similarity between the decoded and presented stimulus orientation), *precision* (the variability of decoded orientations within each bin), and *bias* (the average decoded angle relative to the presented orientation; see *Method – Neural Decoding*).

The results from the neural decoding are shown in cyan in Fig. 3. Accuracy of decoded responses rose sharply from ~50 ms following stimulus onset and gradually reduced over the following 400 ms (Fig. 3a, cyan data), revealing that decodable information is relatively stable across time and robust to additional incoming information, i.e., subsequently presented stimuli, consistent with recent work[23,24]. We found a similar pattern of results for the *precision* of decoded responses (Fig. 3b, cyan data), with precision increasing sharply from ~50 ms and gradually reducing. Precision was significantly above

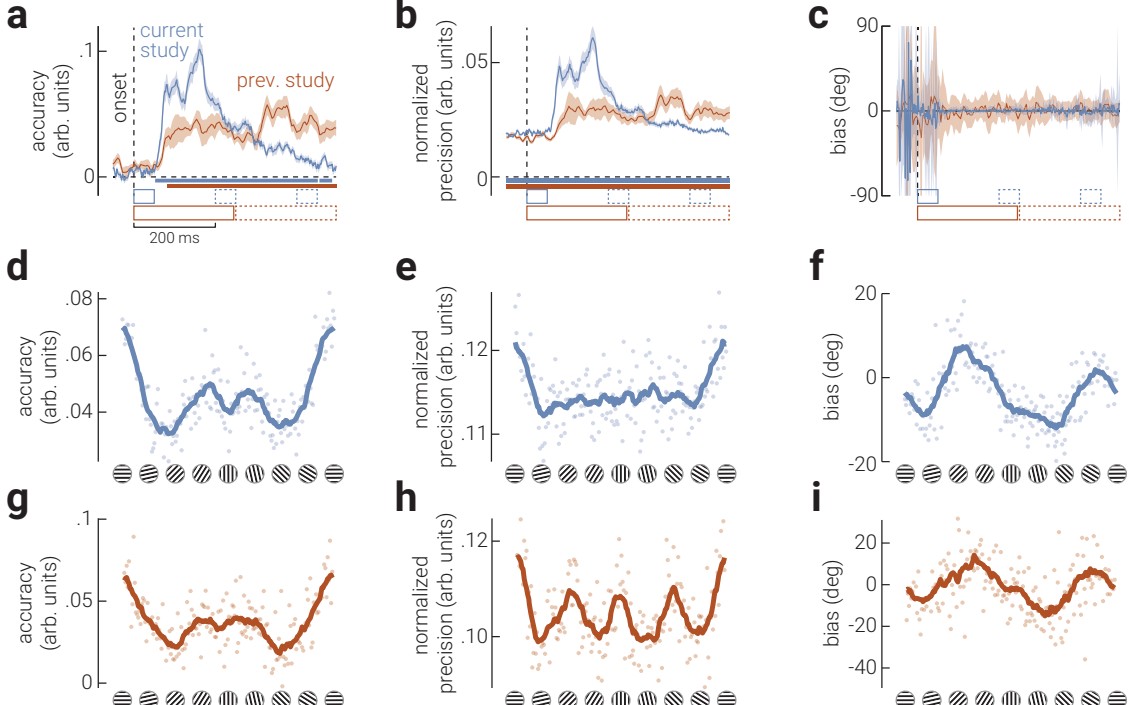

**Fig. 3 | Anisotropies in the neural representation of orientation revealed by inverted encoding.** Average **a**) accuracy, **b**) precision (normalized decoding variability of decoded orientations, from 0 [uniform distribution] to 1 [no variability]), and **c**) bias of orientations decoded from neural responses. Cyan and orange lines indicate data from our experiment and from a previously published study[23], respectively. Shaded error bars in (**a**, **b**) and (**c**) indicate SEM and CI95, respectively; coloured horizontal bars indicate cluster corrected periods that showed a significant difference from chance; horizontal dashed lines indicate chance-level. Note that this difference was significant for the entire period displayed in (**b**). **d**–**f** The time-averaged (**d**) accuracy, (**e**) precision, and (**f**) bias of the inverted model responses, at all orientations, calculated from data collected in our experiment. Cyan lines in (**d**–**f**) indicate moving averages of data points (semi-transparent cyan dots). **g**–**i** Same as (**d**–**f**), but calculated from previously published data[23].

chance prior to stimulus presentation, likely because the decoder produced an over-representation of some orientations, which is consistent with the anisotropic representation of orientations shown in Fig. 2c. As expected, we found no significant deviations in bias when averaged across orientations (Fig. 3c, cyan data). The orange data in Fig. 3 are the results of a re-analysis of a previous study[23], which we will describe further in the following section.

### Neural coding is highly anisotropic

To quantify stimulus-specific variations in neural representations, we measured accuracy, precision, and bias as a function of grating orientation. To do so, we developed a novel inverted model analysis which returns these parameter estimates for each of 180 unique grating stimuli (one for each orientation rounded to the nearest degree; *Methods – Neural Decoding*). To maximize the signal-to-noise ratio for each bin, we averaged results over the period in which there was above-chance accuracy (50–450 ms), but the pattern of results is stable across different periods (Fig. S3).

Neural responses were strongly anisotropic, but not in a way predicted from any leading model of neural coding[15,16]. For accuracy, we found a trimodal pattern of results such that there was one large peak centered on horizontal and two smaller peaks positioned around vertical (Fig. 3d). By contrast, we found a unimodal pattern of results for precision, such that horizontal gratings were decoded most precisely (Fig. 3e). We found a sinusoidal pattern of biases, comprising two attractive curves centered on the cardinal orientations (Fig. 3f). Although we found better accuracy and precision for horizontal orientations, the magnitude of the attractive biases was larger around vertical than horizontal gratings. A split-half analysis of the neural signals in the epoch used to decode orientation parameters confirmed this pattern of results was stable over the course of the epoch (Fig. S3).

To test the reproducibility of these results, we performed these analyses on a separate, previously published, dataset[23]. This prior study differed from our own in many aspects, such as the stimulus parameters, the observers' task, and the temporal design, which resulted in differences in the temporal dynamics of the decoded signal (as shown in Fig. 3a–c, orange data). Despite these considerable differences across experiments, however, we found strikingly similar stimulus-specific variations in neural representations (Fig. 3g–i). Note that the observers' task in the previous study was to report whether grating stimuli were more cardinally or obliquely oriented on a given trial, likely resulting in increased precision around those orientations (Fig. 3h). This re-analysis therefore demonstrates that our neural decoding method is sensitive to task-related goals, with replicable estimates of orientation anisotropies in accuracy and bias.

### Generative modelling of neural population responses

Previous theoretical work provides two primary variations in efficient encoding schemes to explain how anisotropies in orientation coding could be implemented at the neural level[5,15,16]. The proposed neural populations are instantiated such that either neurons tuned to cardinal orientations have narrower tuning (Fig. 4a, uneven tuning width) or that there are more neurons tuned to cardinal orientations (Fig. 4a, uneven tuning preference); however, these competing explanations cannot be resolved using behavioural measurements as they predict the same patterns of estimation error. To adjudicate between these population codes, therefore, we developed a novel generative modelling procedure to simulate EEG activity from neural tuning functions that had either isotropic tuning, or anisotropic tuning as specified by uneven tuning widths or uneven tuning preferences (*Methods – Generative Modelling*). We then applied the same inverted encoding analyses used on the empirical data to estimate accuracy, precision, and bias, as a function of

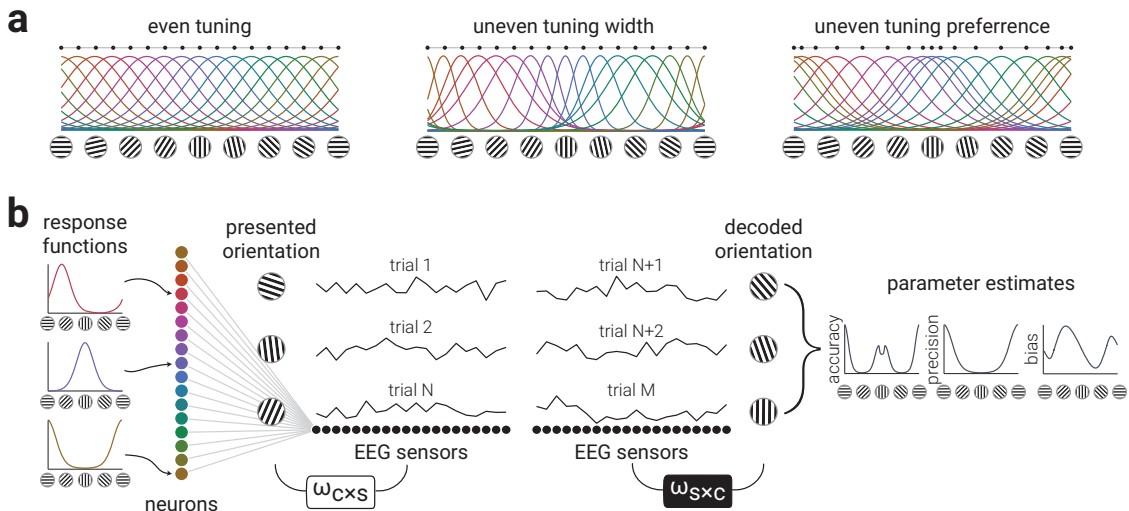

**Fig. 4 | Generative modelling of anisotropic neural tuning properties. a** Neural response functions for a population of orientation tuned neurons. Neurons may have even tuning properties across all orientations (even tuning), but theoretical work suggests that the anisotropies in orientation perception are either explained by denser clustering (uneven tuning preference) or narrower tuning (uneven tuning width) of neurons around cardinal orientations. Horizontally distributed black dots indicate tuning preferences. **b** Illustration of the generative modelling procedure used to simulate neural data to test hypotheses of cardinal biases. A bank of orientation tuned neurons are connected to EEG sensors via random weights. We simulate EEG activity in response to multiple oriented gratings and then decode the orientation using inverted encoding. From the decoded orientations we calculate accuracy, precision, and bias parameters.

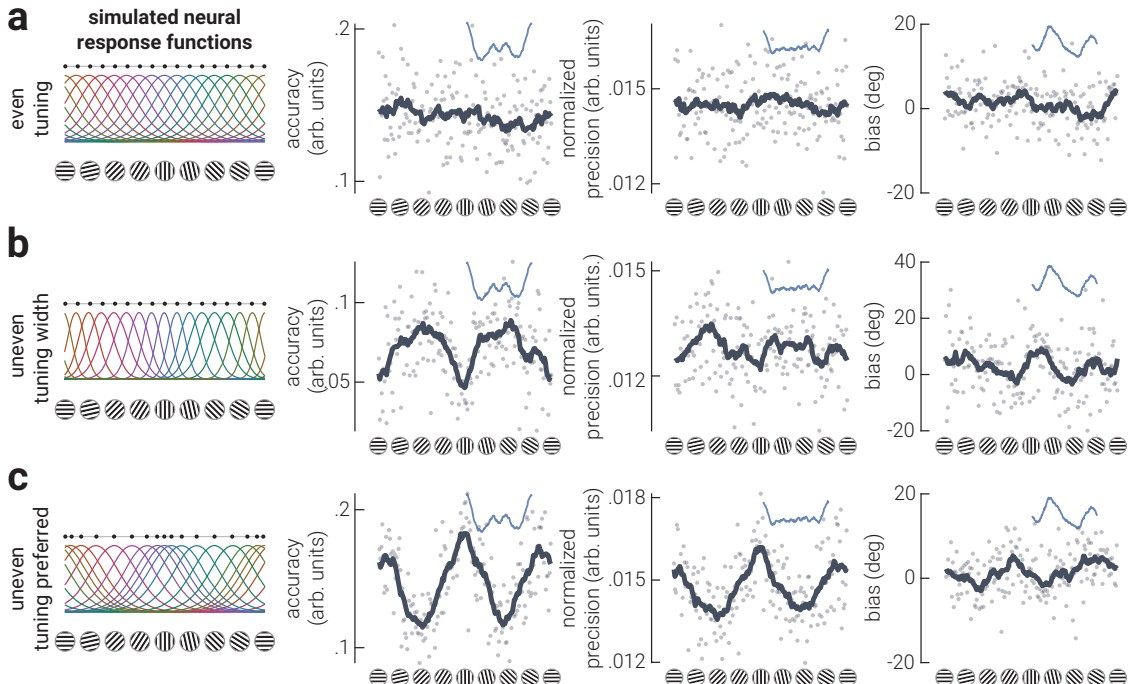

**Fig. 5 | Standard encoding schemes produce anisotropic neural representations of orientation, but do not explain the empirical data. a–c** Results of simulations using neural response functions with (**a**) even or uneven (**b**) width, (**c**) or preference. The left column shows the neural response functions from which data were derived. The columns to the right are the parameter estimate results. Dark grey lines indicate moving averages of data points (semi-transparent dots). Blue insets show the corresponding moving averages from the empirical data; horizontally distributed black dots in the left panels indicate tuning preferences.

orientation (Fig. 4b). While concerns have recently been raised about the extent to which inverted encoding analyses can be used to infer population representations[25–27], our novel generative modelling approach overcomes such issues by directly mapping the transformation of the output of a range of generative models to neural responses. A similar implementation has recently shown that such an approach is sensitive to changes in underlying population representations[28]. If either of the hypothesised models of anisotropic orientation coding explains the true underlying neural implementation, then the empirical data should match the decoded responses of the simulated data from only a single generative population code.

We first generated neural responses from an unbiased isotropic bank of tuning functions (Fig. 4a, even tuning). As expected, this simulation did not replicate the pattern of results observed for the empirical data, as it produced a uniform pattern of accuracy, precision, and biases across all orientations (Fig. 5a). We then generated neural

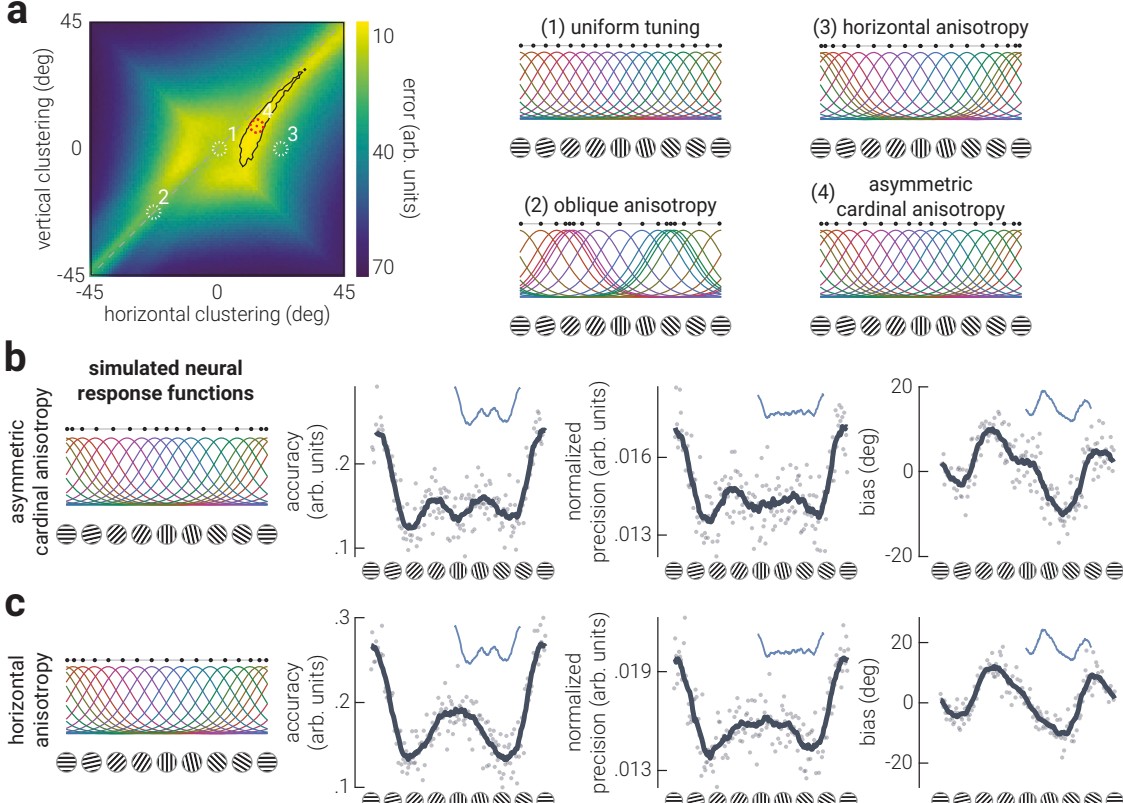

**Fig. 6 | Horizontal preferences explain orientation anisotropies in human visual cortex. a** Difference (error) in discriminability (see *Methods – Generative Modelling* for detailed description) between empirical and simulated data as a function of horizontal and vertical clustering in the generative population code; less error indicates greater similarity between empirical and simulated data. The black outline indicates the region with the lowest 5% of error, and the white dashed circles indicate the position of the example population codes shown on the right. The red dotted circle indicates the position of the best fitting model (4). Results of simulations using neural response functions with (**b**) asymmetric cardinal tuning preferences and (**c**) unimodal horizontal tuning preference. The left column shows the neural response functions from which data were derived. The columns to the right are the parameter estimate results. Dark grey lines indicate moving averages of data points (semitransparent dots). Blue insets show the corresponding moving averages from the empirical data; horizontally distributed black dots in the left panels indicate tuning preferences.

responses from an anisotropic bank of tuning functions specified by narrower tuning around the cardinals (Fig. 5b). Although this population code produced a non-uniform distribution of parameter estimates, the pattern of results did not match the empirical data: accuracy and precision peaked around the obliques, not cardinals, and there was no clear pattern of biases. Finally, we generated neural responses from tuning functions with orientation tuning preferences clustered around the cardinals (Fig. 5c). The pattern of accuracy, precision, and biases produced by this simulation captured the empirical results best out of the three models tested: accuracy and precision peaked around cardinals, with modest attractive biases. In contrast to the empirical results of the two datasets, however, the peak accuracies and biases around horizontal and vertical orientations were equivalent, which was expected from cardinally symmetric response functions, revealing that the modelled response functions did not fully reflect the neural architecture underlying orientation anisotropies in human visual cortex.

Neither of the proposed tuning schemes reproduced the empirical results. We therefore exhaustively searched the space of cardinal and oblique tuning biases to test which population code best explained the empirical data. We chose to manipulate tuning preferences because the uneven tuning preference scheme provided the most promising match to the empirical data (Fig. 5c). The modelled data and empirical data were compared using their *discriminability*[29], a metric that considers the precision and bias – but not the accuracy – of the decoded signal. We computed the similarity between the discriminability of the empirical results and from data generated from population codes comprising all possible combinations of horizontal and vertical biases (Fig. 6a), i.e., by independently varying the clustering around horizontal and vertical orientations. We found that the model that best described the empirical data was a population code with anisotropic cardinal biases, where horizontals are 'more preferred' than verticals by a ratio of approximately 2:1. These results are consistent with our univariate analyses (Fig. 2), and confirm that there is an asymmetry between cardinal orientations, such that horizontals are more preferred than verticals. Further, the ratio of horizontal to vertical asymmetry is consistent with previous estimates of the orientation content in natural visual scenes, but not human-made scenes[5].

A population code with uneven tuning preferences and an asymmetric cardinal anisotropy captured the empirical results well (Fig. 6b): this model reproduced the increased precision at horizontal orientations, and the sinusoidal pattern of biases with a larger attractive bias around vertical gratings. It also reproduced the trimodal distribution of accuracy, despite this metric not being used in the discriminability metric used in the fitting procedure. It is unclear, however, to what extent the redistribution of tuning curves around horizontal and vertical each influence the decoded pattern of anisotropic orientation representation; that is, which changes in accuracy, precision, and biases are related to the horizontal preference, and which are related to the vertical preference? To address this question, we generated EEG data from neural tuning functions with tuning preferences for

horizontal orientations only (Fig. 6c). This simulation captured many of the phenomena observed in the empirical results: it recapitulated the increased accuracy and precision for horizontal gratings, and the asymmetric attractive biases around cardinal gratings. We did not expect a horizontal preference to capture the empirical patterns of results more closely than the classic symmetric cardinal preference models shown in Fig. 5b, c.

A key diagnostic characteristic of the two empirical datasets is the reduction in accuracy for vertical orientations (Fig. 3d, g). This feature is only captured by a population code with asymmetric cardinal preferences (Fig. 6b). To understand why asymmetric cardinal preferences are critical to reproduce the accuracy around vertical orientations, we varied the noise level in the generative model. In the models presented above, we matched each population code to the signal-to-noise of the empirical data by titrating the noise injected into the simulated data (see *Method – Generative Modelling*). By reducing the noise level in the generative models, we can uncover the explanation for the reduction in accuracy around vertical: the reduced accuracy around vertical is due to a small repulsive bias (Fig. S4). This repulsive bias around vertical and the attractive bias around horizontal are additive, resulting in the appearance of a plateau in overall bias around vertical that can be seen in both empirical datasets (Fig. 3f, i), as well as the generated data (Fig. 6b), all of which include the influence of noise.

### Joint coding of the prior and sensory measurement

A general solution to performing Bayesian inference involves embedding the prior *within* the sensory measurement[30–32]. In brief, if tuning curves are distributed such that their sum approximates the log of the prior, then the posterior is a linear readout of the population response, simplifying the biological instantiation of optimal inference (full mathematical detail is given in *Method – Joint coding of the prior and sensory measurement*). We therefore compared how well the properties of the best-fitting population code, derived from our data-driven approach described above, capture environmental priors for natural and constructed scenes (Fig. 7). This analysis revealed that the sensory tuning curves do indeed embed the prior for natural image statistics (left panels, Fig. 7). The activity of the population code that best captures the responses of the human visual system therefore provides a stimulus representation from which a full (log) posterior can be computed through simple linear summation of activity, resulting in optimal inference in natural scenes. By comparison, the properties of the population code provide a poorer description of the prior for constructed scenes (right panels, Fig. 7).

## Discussion

We investigated whether the human visual system represents environmental attributes that are critical for optimal inference. Using our novel

decoding analyses, we quantified surprising anisotropic neural responses that deviated from the expectations of leading models of visual prediction. By developing novel generative modelling tools, we were able to recover the underlying tuning properties of the population code driving the anisotropic neural responses. In addition to the cardinal anisotropy of orientation coding suggested by prior computational and behavioural investigations[5,15,16], we reveal that an asymmetry in cardinal representations is an inherent property of visual coding. Most importantly, our data provide the first clear evidence for a recent hypothesis about how priors are instantiated in biological systems[30]: the prior is embedded in the sensory measurement (Fig. 7). By adjusting tuning curves to prioritise horizontal orientations and, to a lesser extent, vertical orientations, the early visual system implicitly combines the prior for natural images with incoming sensory signals. This coding scheme negates the need for the prior to be represented by a separate population of neurons, thereby improving neural efficiency.

The prioritisation of neural resources to horizontal orientations can be understood in the context of efficient coding[33]. A neural code is efficient if resources are allocated proportionally to the variation in the input signal along some feature dimension. While most prominent computational work models the anisotropic distribution of orientations as being symmetrical across cardinals, differences in horizontal and vertical contrast energy have been reported in measurements of natural image statistics[6,7] and a matching asymmetry has been observed in the tuning preferences of orientation-selective neurons[9–11]. For example, in a recent analysis of over 26,000 high quality digital photos of natural images, Harrison[7] reported that horizontal orientations were 10% more prevalent relative to vertical orientations. In an earlier analysis of 60 images, Hansen and Essock[20] found a similar over-representation of horizontal information across multiple spatial scales, suggesting the dominance of horizontal information is a scale invariant property of natural scenes. In their Bayesian account of visual orientation perception, Girshick et al.[5] derived estimates of observers' priors for orientation and found a close correspondence to cardinally-biased, but symmetric, environmental statistics. Close inspection of their results, however, shows that both the average of observers' recovered prior and the distribution of image statistics indicated greater representation of horizontal over vertical information, particularly in images of natural scenes (their data are re-plotted as the coloured traces in Fig. 7). A sensory population code that prioritises horizontal information over vertical, and cardinals over obliques, can thus be interpreted as representing the true distribution of environmental statistics derived from natural scenes. It is interesting to note that the sum of the tuning curves provides a relatively poor approximation of the statistics of constructed scenes, raising the possibility that the default tuning of sensory systems depends more on statistics accumulated over evolutionary time scales than developmental time scales.

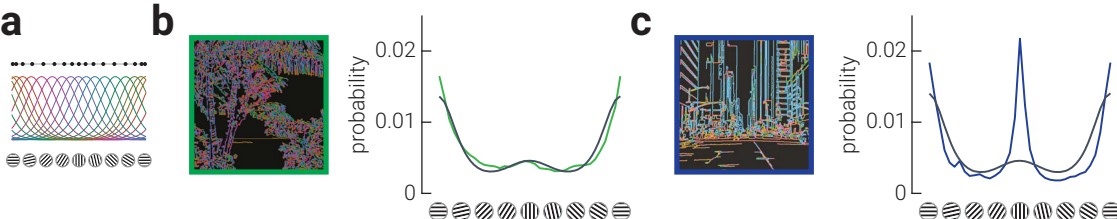

**Fig. 7 | The redistribution of tuning curves effectively embeds a prior for natural image statistics in the sensory measurement. a** Horizontal-biased population code that best explained the neural responses, as described in Results. **b, c** The left and right plots show the match of the properties of this population code with the distribution of orientation statistics for natural scenes (**b**) and constructed scenes (**c**) as measured by Girshick et al. (2011). In each plot, the coloured trace shows the probability of each orientation in a set of images, while the black trace shows the exponentiated sum of the best fitting tuning functions recovered from

our generative model. Extending theory developed by Simoncelli (2009), the exponentiated sum of the tuning curves is proportional to the prior for natural scenes: $\exp(\sum f_n(\theta)) \propto P(\theta)$ Assuming Poisson neural variability, this allows the log posterior to be obtained as a weighted sum of the neural responses: $\log P(\theta|r) = \sum r_n \log f_n(\theta) + const.$, providing an efficient means by which to perform optimal inference in the human visual system. Beach photo credit: Matthew Brodeur, https://unsplash.com/photos/DH_u2aV3nGM. City street photo credit: Andrea Cau, https://unsplash.com/photos/nV7GJmSq3zc.

Linking anisotropies in orientation perception to neural tuning schemes is challenging because behavioural measurements are confounded by response biases, which can be in direct conflict with perceptual biases[34]. Here we avoided this confound by measuring biases in the neural representation of passively viewed oriented gratings, and we then replicated these effects using an independent dataset in which observers were actively engaging in a orientation discrimination task. A further advantage of our method is that it allowed us to directly test, through generative modelling, the population level neural tuning properties that give rise to anisotropies in the representation of orientation. The impact of this method is highlighted by the inability of previous work to adjudicate between population codes designed with uneven tuning preferences versus uneven tuning widths. Our results may appear to contradict earlier neuroimaging studies that found horizontal and vertical orientations are equally over-represented in cortex relative to obliques[35]. However, close inspection of these earlier data also reveals a horizontal/vertical asymmetry. It is therefore likely that the assumption of equal cardinal responsiveness has led the asymmetries we found to have been overlooked in more recent neuroimaging investigations[36,37].

Beyond providing a diagnostic marker for the underlying neural implementation of orientation tuning in the human visual cortex, the reduced accuracy and small repulsive bias around vertical gratings (Fig. S4) may explain previous inconsistencies in the behavioural literature of perceptual cardinal biases. In particular, both attractive and repulsive perceptual biases have been reported around the cardinal orientations, when measured using different experimental paradigms[5,34,38]. This discrepancy has been explained within a Bayesian framework as the result of either manipulating external (stimulus) or internal (neural) noise[16]. Under this framework, humans are assumed to have a prior distribution that gives preference to cardinal orientations. Increasing external noise results in a greater reliance on this prior (attraction towards cardinals), whereas increasing internal noise produces a bias away from this prior (repulsion away from cardinals). Our findings show that there is a small repulsive bias around vertical, which is surrounded by a - considerably larger - attractive bias. It is feasible that the smaller repulsive bias could be obscured by the larger attractive bias under conditions of low signal-to-noise. Thus, this provides an alternative explanation for how manipulating noise could lead to observing either attractive or repulsive biases (at least around vertical orientations), which can be tested in future psychophysical investigations.

The neural responses recorded in the current study with EEG represent activity from primary visual cortex in addition to other upstream visual areas (e.g., V2). Thus, while we modelled the orientation tuning functions of neurons based on the known properties of V1 neurons, it is likely that other neurons within the visual processing cascade, which have more complex tuning properties, also contributed to the recorded representation. While neurophysiological work suggests similar orientation biases are present in higher visual regions, e.g., V5/MT[39], future work using an imaging technique with better spatial specificity, e.g., fMRI, could test whether the same prior is instantiated across different cortical areas.

In summary, here we apply novel neural decoding and generative modelling approaches that reveal, replicate, and explain hitherto unknown aspects of sensory anisotropies. For the first time, we adjudicate between competing models of uneven neural tuning, demonstrating that uneven tuning preferences, but not tuning width, explain cardinal biases. We further show that anisotropies in the representation of orientation are almost entirely explained by a preference for horizontal orientations, while the smaller preference for vertical orientations produces a small repulsive bias that may account for previous discrepancies in attractive and repulsive cardinal bias. More broadly, we demonstrate both a new neural decoding algorithm and a corresponding generative modelling procedure that can be used in a principled manner to describe neural representations in high fidelity and explain their underlying architecture.

## Methods

### Participants

Thirty-seven neurotypical human adults (mean ± standard deviation age, 23.8 ± 4.6 years; 23 females) participated in the experiment. Observers were recruited from The University of Queensland, had normal or corrected-to-normal vision (assessed using a standard Snellen eye chart), and were required to pass an initial screening session to qualify for the experiment (see *Stimuli, task, and procedure* Section for details). Data from one participant were omitted from analyses due to hardware failure. All participants were naïve to the aims of the experiment and gave informed written consent. The experiment was approved by The University of Queensland Human Research Ethics Committee.

### Apparatus

The experiment was conducted in a dark, acoustically and electromagnetically shielded room. The stimuli were presented on a 24-inch ViewPixx monitor (VPixx technologies, Inc., Montreal) with 1920 × 1080 resolution and a refresh rate of 144 Hz. Viewing distance was maintained at 45 cm using a chinrest, meaning the screen subtended 61.18° × 36.87° (each pixel 2.4' × 2.4'). Stimuli were generated in MATLAB v2020a (The MathWorks, Inc., Matick, MA) using Psychophysics Toolbox[40,41] v3.0.18.13 (see http://psychtoolbox.org/). EEG signals were recorded using 64 Ag-AgCl electrodes (BioSemi, Amsterdam, Netherlands).

### Stimuli, task, and procedure

The stimuli comprised sinewave gratings (1 cycle/°, 0.5 contrast, random phase) presented centrally within a circular aperture (radius 4.2°), which was smoothed at the edges, on a mid-grey background. A centrally positioned green fixation dot (radius 0.25°) was presented to reduce eye movements. To maintain attention, participants were instructed to count the number of 'target' stimuli, in which the spatial frequency of the grating was reduced (0.66 cycle/°). Between 0 and 3 targets appeared during each trial, selected at random.

Trials consisted of grating stimuli (orientations randomly selected between 0 and 180°) presented for .05 s each, separated by a blank 0.15 s interstimulus-interval (ISI) for 10 s (neural probe). The numbers 0–3 were then displayed on the screen and participants were given 2 s to indicate the number of targets presented, using the mouse (detection task). This was repeated 12 times per block. Participants performed 6 blocks of trials (~20 min), receiving feedback on their detection accuracy at the end of each block.

Prior to the main experiment, participants were required to pass an initial screening session in which they completed the same task but without EEG. To pass the screening, participants were required to perform above chance (25%) on the detection task. Further, they were required to show a significant repulsive bias on a separate orientation reproduction task (assessed using a one-tailed *t*-test), which was related to data collected for a different study. This resulted in approximately half of the participants being screened out (32 due to insufficient repulsive bias, and 8 due to both insufficient repulsive bias and low detection task performance). Screening and experiment sessions were separated by a minimum of 24 h. Note that the individuals who were screened out are not included in the *Participants* section.

### EEG

The EEG signals were digitised at 1024 Hz sampling rate with a 24-bit A/D conversion. The 64 active scalp Ag/AgCl electrodes were arranged according to the international standard 10–20 system for electrode placement[42] using a nylon head cap. As per BioSemi system design, the common mode sense and driven right leg electrodes

served as the ground, and all scalp electrodes were referenced to the common mode sense during recording. Offline EEG pre-processing was performed using EEGLAB v2021.1[45] in accordance with best practice procedures[43,44]. The data were initially down-sampled to 512 Hz and subjected to a 0.5 Hz high-pass filter to remove slow baseline drifts. Using EEGLAB, electrical line noise was removed using *pop_cleanline.m with the following parameters: bandwidth = 2; all channels; linefreqs = [50 100 150 200 250]; normSpectrum = 0; p = 0.01; pad = 2; scanforlines=true; sigtype=channels; tau = 100; winsize = 4; winstep = 4*. The *clean_artifacts.m*[45] function (default parameters) was used to remove bad channels (identified using Artifact Subspace Reconstruction), which were then spherically interpolated from the neighbouring electrodes. Only individual channels were removed and interpolated, no time periods were removed from the data. Data were then re-referenced to the common average before being epoched into segments around each neural probe stimulus (−0.25 s to 0.5 s from the stimulus onset). Systematic artefacts from eye blinks, movements and muscle activity were identified using semi-automated procedures in the SASICA toolbox[46] and regressed out of the signal.

### Neural decoding

To characterise sensory representations of the stimuli, we used an inverted modelling approach to reconstruct the orientation of the gratings from the EEG signal[22]. We make several advances in the application of our models that negate the recent concerns[25,26] about the validity of the properties of the recovered population code (see *Generative Modelling* section below). A theoretical (forward) model was nominated that described the measured activity in the EEG sensors given the orientation of the presented grating. The forward model was then used to obtain the inverse model that described the transformation from EEG sensor activity to stimulus orientation. The forward and inverse models were obtained using a ten-fold cross-validation approach in which 90% of the neural probe data were used to obtain the inverse model on which the remaining 10% were decoded. In our modelling, we assume that EEG sensor noise is isotropic across orientations and additive with the signal; while this assumption would be violated if there were a systematic relationship between orientation preference and anatomical location across participants, abundant evidence rules out such a relationship in human and non-human brains[9,47–49].

Similar to previous work[21], the forward model comprised six hypothetical channels, with evenly distributed idealized orientation preferences between 0° and 180°. Each channel consisted of a half-wave rectified sinusoid raised to the fifth power. The channels were arranged such that a tuning curve of any orientation preference could be expressed as a weighted sum of the six channels. The observed EEG activity for each presentation could be described by the following linear model:

$$\mathbf{B} = \mathbf{WC} + \mathbf{E} \tag{1}$$

where **B** indicates the ($m$ sensors × $n$ presentations) EEG data, **W** is a weight matrix ($m$ sensors × 6 channels) that describes the transformation from EEG activity to stimulus orientation, **C** denotes the hypothesized channel activities (6 channels × $n$ presentations), and **E** indicates the residual errors.

To compute the inverse model, we estimated the weights that, when applied to the data, would reconstruct the underlying channel activities with the least error. In line with previous magnetencephalography work[24,50], when computing the inverse model, we deviated from the forward model proposed by[21] by taking the noise covariance into account to optimize it for EEG data, given the high correlations between neighbouring sensors. We then estimated the weights that, when applied to the data, would reconstruct the underlying channel

activities with the least error. Specifically, **B** and **C** were demeaned such that their average over presentations equalled zero for each sensor and channel, respectively. The inverse model was then estimated using a subset of the data, selected through cross-fold validation (10 folds). The hypothetical responses of each of the six channels were calculated from the training data in each fold, resulting in the response row vector $\mathbf{c}_{train,i}$ of length $n_{train}$ presentations for each channel $i$. The weights on the sensors $\mathbf{w}_i$ were then obtained through least squares estimation for each channel:

$$\mathbf{w}_i = B_{train}\mathbf{c}_{train,i}^{\mathrm{T}}\left(\mathbf{c}_{train,i}\mathbf{c}_{train,i}^{\mathrm{T}}\right)^{-1} \tag{2}$$

where $\boldsymbol{B}_{train}$ indicates the ($m$ sensors × $n_{train}$ presentations) training EEG data. Subsequently, the optimal spatial filter $\mathbf{v}_i$ to recover the activity of the $i$th channel was obtained as follows[50]:

$$\mathbf{v}_i = \frac{\widetilde{\Sigma}_i^{-1}\mathbf{w}_i}{\mathbf{w}_i^{\mathrm{T}}\widetilde{\Sigma}_i^{-1}\mathbf{w}_i} \tag{3}$$

where $\widetilde{\Sigma}_i$ is the regularized covariance matrix for channel $i$. Incorporating the noise covariance in the filter estimation leads to the suppression of noise that arises from correlations between sensors. The noise covariance was estimated as follows:

$$\widehat{\Sigma}_i = \frac{1}{n_{train}-1}\boldsymbol{\varepsilon}_i\boldsymbol{\varepsilon}_i^{\mathrm{T}} \tag{4}$$

$$\boldsymbol{\varepsilon}_i = \mathbf{B}_{train} - \mathbf{w}_i\mathbf{c}_{train,i} \tag{5}$$

where $n_{train}$ is the number of training presentations. For optimal noise suppression, we improved this estimation by means of regularization by shrinkage using the analytically determined optimal shrinkage parameter[50], yielding the regularized covariance matrix $\widetilde{\Sigma}_i$.

In the orientation focused analyses (Fig. 3d, e), we adapted a method recently introduced by[27], which uses sparsely distributed channels to produce a dense representation of channel responses. The standard inverted modelling approach typically involves training and testing a model using a relatively small number of channels (e.g., 6) to represent the entire spectrum of identities across a given feature (e.g., orientation). The 'enhanced inverted encoding model' (eIEM) method involves repeatedly computing the inverse model, while rotating the forward model channel orientation preferences until each orientation is represented by a channel response (e.g., 180 channels, one for each orientation). The eIEM method was established using data with feature identities sampled from discrete bins (e.g., 0°, 30°, 60°, 90°, 120° and 150°), which determine: a) the number of channels used in each model (e.g., 6), and b) the number of feature identities from which predictions could be derived from the dense output (e.g., 6). By contrast, we presented observers with orientations sampled from a continuous uniform distribution of all orientations. Thus, we first sorted presentations into five orientation bins (0°, 36°, 72°, 108°, and 144°). After rotating channels to produce 180 channel responses for each presentation, we then binned the responses according to the original grating orientation (rounded to nearest the nearest degree); that is, the orientation of the presentation prior to binning. This allowed us to produce high-resolution model predictions for each orientation. We performed this analysis at each time point within the epoch, but to improve the signal of the estimates we averaged predictions across time points when there was reliable decoding accuracy (50–450 ms following stimulus onset).

For each presentation, we decoded orientation by converting the channel responses to polar form:

$$z = \mathbf{c} \cdot e^{2i\boldsymbol{\varphi}} \tag{6}$$

and calculating the estimated angle:

$$\hat{\theta} = \frac{\arg(z)}{2} \tag{7}$$

where $\mathbf{c}$ is a vector of channel responses and $\boldsymbol{\varphi}$ is the vector of angles at which the channels peak (multiplied by two to project 180° orientation space onto the full 360° space). From the decoded orientations, we computed three estimates: *accuracy*, *precision*, and *bias*. Accuracy represented the similarity of the decoded orientation to the presented orientation[24], and was expressed by projecting the mean resultant (averaged across presentations of the same grating orientation) of the difference between decoded and grating orientations onto a vector with 0°:

$$\hat{r}_\theta = \mathrm{Re}[\bar{R}], \bar{R} = \frac{1}{n}\sum_{j=1}^{n}\exp\left(\mathrm{i}\left(\hat{\theta}_j - \theta\right)\right) \tag{8}$$

Precision was estimated by calculating the angular deviation[51] of the decoded orientations within each orientation bin:

$$\hat{\sigma}_\theta = \sqrt{2\left(1 - |\bar{R}|\right)} \tag{9}$$

and normalized, such that values ranged from 0 to 1, where 0 indicates a uniform distribution of decoded orientations across all orientations (i.e., chance-level decoding) and 1 represents perfect consensus among decoded orientations:

$$\hat{p}_\theta = 1 - \frac{2\hat{\sigma}_\theta}{\sqrt{2}} \tag{10}$$

Bias was estimated by computing the circular mean of angular difference between the decoded and presented orientation:

$$\hat{b}_\theta = \arg(\bar{R}) \tag{11}$$

Prior to the main neural decoding analyses, we established the sensors that contained the most orientation information by treating time as the decoding dimension and obtaining inverse models for each sensor, using 10-fold cross-validation. This analysis revealed that orientation was primarily represented in posterior sensors (Fig. S2); thus, for all subsequent analyses we only included signals from the parietal, parietal-occipital, occipital, and inion sensors to compute the inverse model.

### Generative modelling

To characterise changes in neural tuning which might have given rise to the empirical results observed, we used inverted modelling to decode orientation from simulated EEG data produced by neural populations with either isotropic tuning or anisotropic tuning specified by differences in tuning preferences or tuning widths. Although the empirical EEG data comprises a time series of neural activity, we only simulated temporally static data as we were comparing this with the time averaged empirical data. Simulated data were produced by assuming variable weights between a bank of response curves, each of which represented the aggregate responses of neural populations with similar tuning, with different orientation preference ($n = 16$) and EEG sensors ($m = 32$). As with the forward model, the neural functions were arranged such that a tuning curve of any orientation preference could be expressed as a weighted sum of the 16 functions. Thus, for each simulated presentation, the EEG activity was computed as:

$$s_i = \mathbf{c}_\theta \mathbf{w}_i + u \tag{12}$$

where $s_i$ denotes the activity of sensor $i$, $\mathbf{c}_\theta$ indicates the set of tuning curves evaluated at orientation θ, $\mathbf{w}_i$ denotes the weights between the sensor $i$ and the response functions, and $u$ indicates Gaussian noise (s.d. = 6). The noise level was selected to approximate the level of noise observed in the empirical results. Neural tuning consisted of von Mises functions ($\kappa = 2$), normalized such that they ranged from 0 to 1. Neuron-to-sensor weights ($\mathbf{w}_i$) were randomly assigned from a uniform distribution between 0 and 1. This range was selected for simplicity. The variance in weights between the activity of subpopulations of neurons and EEG sensors in the brain is likely different, however, our simulations confirmed that modifying this range had no qualitative influence on the pattern of results. Uneven *tuning preferences* were modelled by shifting the neural tuning functions according to the sum of two von Mises derivative functions ($\kappa = 0.5$) centered on 0° (amplitude = 15) and 90° (amplitude = 10), such that the maximum shift applied was 15°. This resulted in shifting the neural response functions towards the cardinals, with increased clustering around horizontal relative to vertical. Uneven *tuning widths* were modelled by narrowing the neural tuning functions according to the sum of two von Mises functions ($\kappa = 2$) centered on 0° (amplitude = 15) and 90° (amplitude = 10), such that the maximum increase to $\kappa$ was 15. This resulted in narrowing the neural response functions that preferred cardinal orientations, with increased narrowing around horizontal relative to vertical.

In line with the empirical experiment, the orientation of each simulated presentation was drawn from a uniform distribution between 1 and 180°. We simulated 3600 presentations from the even and uneven neural response functions. We then applied the same analyses as used on the empirical data to estimate accuracy, precision, and bias as a function of orientation. The final presented results are the average of parameters estimated from 36 simulated datasets.

To find the orientation anisotropy that best captured the empirical neural data, we computed the difference (error) between the empirical data and data that was simulated using all possible combinations of cardinal and oblique neural tuning preferences. The error was calculated as the difference between the *discriminability* of the decoded orientations from empirical and simulated data. Discriminability combines measures of precision and bias to provide an index of how discriminable signals are at each orientation[29]:

$$d_\theta = \frac{\hat{\sigma}_\theta}{1 + \hat{b}_\theta'} \tag{13}$$

where $\hat{b}_\theta'$ is the change in bias at orientation θ, which we estimated as:

$$\hat{b}_\theta' \approx \frac{\hat{b}\left(\theta + \frac{\delta}{2}\right) - \hat{b}\left(\theta - \frac{\delta}{2}\right)}{\delta} \tag{14}$$

where δ indicates the size to the window of estimation (16°). The generative data was stochastic, due to the injected noise, so we calculated the error ten times for each model and used the average. Neural tuning preference anisotropies were produced in the same manner as described above, and the amplitude of the shift was independently varied around both horizontal (0°) and vertical (90°) orientations, from −45° to 45°, in increments of 1°, resulting in a total of $91 \times 91 = 8281$ models.

### Joint coding of the prior and sensory measurement

Although there is general agreement that the brain combines prior information with sensory signals, the physiological implementation of such Bayesian inference is less clear[52–55]. In a population of $N$ neurons

with independent Poisson variability and tuning curves $f(\theta)$, the joint likelihood of the response of the population given a stimulus of orientation $\theta$ is:

$$P(\mathbf{r}|\theta) = \prod_{n}^{N} \frac{f_n(\theta)^{r_n}}{r_n!} e^{-f_n(\theta)} \qquad (15)$$

where $\mathbf{r}$ is a vector containing each unit's spike count and $r_n$ is the number of spikes for unit $n$. It is convenient for biological sensory systems to signal log probabilities so that joint (log) probabilities can be computed linearly, for example by summing spikes:

$$\log P(\mathbf{r}|\theta) = \sum_{n}^{N} r_n \log f_n(\theta) - \sum_{n}^{N} f_n(\theta) - \sum_{n}^{N} \log r_n! \qquad (16)$$

Bayes' rule tells us that the posterior is computed by multiplying the likelihood, for example as computed above, with a prior, $P(\theta)$, and dividing by the marginal likelihood, $P(\mathbf{r})$. Using log probabilities, the formulation for computing the log-posterior can then be written as:

$$\log P(\theta|\mathbf{r}) = \sum_{n}^{N} r_n \log f_n(\theta) - \sum_{n}^{N} f_n(\theta) + \log P(\theta) + C_1 \qquad (17)$$

where $C_1$ is a normalizing constant that does not depend on $\theta$ and can be ignored for purposes of comparing posterior probabilities or obtaining a *maximum a posteriori* (MAP) estimate. Estimating the posterior in this way would require at least three populations of neurons (or if, for example, the computations are performed via the weights of lateral connections, three sets of weights): one set to encode the stimulus, one to represent the prior, and another to decode the signal. As noted by Simoncelli[30], a more general solution to performing Bayesian inference involves embedding the prior into the sensory measurement by distributing the sensory tuning curves such that their sum is equal (up to an additive constant) to the log of the prior:

$$\sum_{n}^{N} f_n(\theta) = \log P(\theta) + C_2 \qquad (18)$$

In this case, the second and third terms of Eq. 13 cancel out, such that the full log-posterior distribution can be computed (up to an additive constant) as a simple linear function of the sensory measurement:

$$\log P(\theta|\mathbf{r}) = \sum_{n}^{N} r_n \log f_n(\theta) + C_3 \qquad (19)$$

This solution is more efficient than encoding the prior in a separate population, requiring only two populations: one to encode the stimulus and another to obtain the linear readout over the population.

The equations above specify optimal inference for a population of spiking neurons, whereas we measured meso-scale activity across the scalp. On the assumption that the best fitting population code we recovered from the neural data is a linear transformation of the underlying neural activity, then if the prior is embedded in the sensory measurement Eq. 18 should hold. Or equivalently,

$$e^{\sum_{n}^{N} f_n(\theta)} \propto P(\theta) \qquad (20)$$

To test this, we fit the exponentiated sum of the best-fitting tuning curves in our study to the distribution of image statistics as measured by Girshick et al.[5] for natural scenes versus constructed scenes using two separate linear models. While the sum of the tuning curves predicted both sets of image statistics ($ps < .001$), they almost perfectly

recovered the statistics of natural scenes, accounting for 92% of the variation in natural scene statistics, while providing a poor fit to constructed scenes, accounting for only 46% of the variation in constructed scene statistics. The requirement that the exponentiated sum of the tuning curves is proportional to the prior, as specified by Eq. 20, is therefore met for natural image statistics. The recovered coding scheme of the human visual system thus embeds the prior in the sensory measurement, providing a concise means by which to perform optimal inference.

### Re-analysis of published data
To test the reproducibility of our empirical results, we re-analyzed a dataset from a comparable, previously published, study from another laboratory[23]. This dataset comprised 32 channel EEG activity in response to viewing of uniformly sampled oriented gratings. In King and Wyart's[23] experiment, observers viewed eight rapidly presented (5 Hz) oriented gratings per trial and were tasked with reporting whether the gratings were, on average, aligned more to the cardinal or oblique axes. In total, there were approximately 4800 presentations per participant, with 15 participants included; for a more detailed description of the experiment and data, see[23]. We received the pre-processed data from the authors, on which we performed the same inverted encoding analyses as used on the data from our experiment.

### Statistical analyses
Statistical analyses were performed in MATLAB v2020a and CircStat Toolbox v1.12.0.0[56]. For analyses of differences in univariate responses and parameter estimates between orientations as a function of time, a cluster correction was applied to remove spurious significant differences. First, at each time point, the effect size of orientation was calculated. A repeated measures analysis of variance was applied to calculate the $F$ statistic associated with orientation. Next, we calculated the summed value of these statistics (separately for positive and negative values) within contiguous temporal clusters of significant values. We then simulated the null distribution of the maximum summed cluster values using permutation ($n = 1000$) of the orientation labels, from which we derived the 95% percentile threshold value. Clusters identified in the data with a summed effect-size value less than the threshold were considered spurious and removed.

### Reporting summary
Further information on research design is available in the Nature Portfolio Reporting Summary linked to this article.

## Data availability
The EEG data generated in this study have been deposited in the following OSF database: https://osf.io/5ba9y/. Source data are provided as a Source Data file. Source data are provided with this paper.

## Code availability
Code has been deposited in GitHub (https://github.com/ReubenRideaux/Neural-tuning-instantiates-prior-expectations-in-the-human-visual-system).

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

## Acknowledgements
We thank Valentin Wyart and Jean-Remi King for sharing their data. This work was supported by an Australian Research Council Discovery Early Career Researcher Award to RR (DE210100790) and Wellcome Trust Senior Fellowship to PMB (106926).

## Author contributions
R.R. contributed to conceptualization, methodology, software, formal analysis, investigation, data collection, data curation, visualization, project administration, and writing – original draft and revisions. W.J.H. contributed to conceptualization, methodology, investigation, and writing – original draft and revisions. P.M.B. contributed to methodology and writing – revisions.

## Competing interests
The authors declare no competing interests.
