## [Peer Review File · Nature Communications]

Neural tuning instantiates prior expectations in the human visual systemREVIEWER COMMENTS

Reviewer #1 (Remarks to the Author):

Review of "Neural tuning instantiates prior expectations in the human visual system"

This paper explores the coding of orientation in the brain using a forward model (similar to Brouwer/Heeger) and varies the population parameters to fit the data, with the best-fitting model having a denser collection of tuning curves at the cardinal orientations, with unequal skewing of the density for horizontal vs. vertical. I was fairly amazed that this could work using EEG data (I have no experience with that).

I'm also a bit confused by what the result means in terms of cortical coding, since the collection of decoded electrodes presumably spanned all of early visual cortical regions and beyond. When folks first started decoding orientation with fMRI in V1 (Frank Tong), folks on my floor complained that the reason it worked was not that voxels have slight random biases toward one orientation column or another, but rather that there was an overall regional bias depending on polar angle in the retinotopic map (Freeman/Merriam/Heeger), and that debate continues to this day. You are using EEG signals, which pool over a far broader region per sensor, so sure, you can decode, but why can you do so and why does it make sense to relate those signals to a channels model based on tuning in V1 alone?

Regardless, this is an interesting paper and a real contribution.

Comments by line number:

81: "from the grand average of all gratings" - I suppose this should be clarified as "from the grand average of responses, averaged over responses to all gratings" or some such. The same applies to line 94.

117: Looking for bias averaged over orientations seems a bit silly, eh?

Figure 3h looks a lot different from 3e and those three peaks are kind of bizarre, yes? Nevertheless, these results from extant data (the replication) are quite impressive.

210: responses and biases ... were equivalent: That was guaranteed as your model was symmetric in that way.

351: At this point you haven't yet pointed out that, although the usual Bayesian setup suggests attraction to the cardinals, that behavior in estimation tasks (unlike my discrimination task with Girshick and Simoncelli) often finds repulsion from the cardinals. Your discussion on Wei/Stocker on this seems a bit thin, since you have a setup that guarantees relatively symmetric likelihoods, and thus your model won't produce the repulsion in estimates that Wei/Stocker think is correct and, in their cost function, optimal. You do talk about this on the next page, and you do get this from your asymmetric model, but I don't know that the literature finds asymmetries with attraction to horizontal and repulsion from vertical or vice versa (although I thought Frank Durgin once told me that was a finding and folks related it to something about ground planes in the environment ... whatever...).

429: Hmm, I'd think that more text should be devoted to this massive drop in participants you analyzed.

474: I would have liked to see more text here about this analysis method and, in particular, how you get enough data to estimate a covariance matrix (which usually needs more data than one can gather to get a stable estimate).

487-497: I found this text to be opaque.

Eq. 2: The equation is a bit of a shorthand since c is a vector and you are like the $\exp()$ is also producing a vector, so that this is a complex dot product, yes? Makes sense, but it's a shorthand that might trip people up. In any case, z is a complex number with a length that truly is a kind of response strength, but it isn't what you use. Instead, you use \hat{r}_θ . This latter value (Eq. 4) isn't a strength per se. Rather, you average (you need to state what trials you are averaging over) unit complex difference vectors and ask what is the component in the "correct" direction and call that a response strength. That seems a misnomer to me. Much of this page is standard circular statistics (adapted to the range 0 - π) and could use a little clarification for those who are unfamiliar with this.

532: I'm not sure what "the tuning curve of orientation θ " means, but c_θ is, rather, the set of tuning curves evaluated at orientation θ .

535-542: This was a bit unclear as well; be kind to your readers ;^)

Reviewer #2 (Remarks to the Author):

Thank you for the opportunity to review this paper. Wonderful to see EMCRs conducting work of this quality. Its coupling of several technically sophisticated analytical methods allowed the authors to uniquely identify the specific form of orientation anisotropy instantiated in the human visual cortex. Viewed in isolation, the contribution of this article appears narrow, in that it merely resolves the disputed parameterization of orientation biases well-known to exist in human visual processing. But this is the wrong lens with which to view this data, because while it largely answers these outstanding questions, its novel methodology also answers all the intervening questions and thereby presents methods and insights likely useful beyond psychophysics and cognitive neuroscientists. These insights are afforded by viewing and quantifying the data using cumulatively novel modelling approaches. The visual communication of these methods and their findings with clean, neat figures was particularly commendable, as it demonstrates the authors' novel perspective on this issue immediately. The observation that humans have a vertical-biased, cardinal orientation preference which aligns with what is observed in decomposed of natural scenes is an insight useful to those working on inferential statistics, computer vision and evolutionary biology. Our visual system acts as if it possesses the same biases as would be needed in an efficient Bayesian estimator trained to process natural scenes, but not scenes of constructed architecture. This is a fundamentally interesting observation.

The work generally appears sufficient to support the claims. The management of the EEG data appears sound, although more could have been done to explain the precise transformations and cleaning methods used. What was the ERP baseline? What parameters were used in the artifact cleaning toolboxes called? What kind of rejection and interpolation was used (per channel and in ASR)? In my experience, these parameters can strongly impact on the data patterns that emerge. The choice of analysis window, while data-driven, was slightly unexpected. The quantification window (50-400ms) runs into the presentation of the subsequent stimulus, and so too does the baseline window (-250ms), although I did not see any evidence that baselining took place (Fig 2b also implies some variation in ERP prior to stimulus onset). Given the caution evident in the authors' choice of time window and electrode set (Figures S1 and S2 were welcome additions), which exceeds standards in the field, I am confident there are principled reasons for this design choice but the article would be strengthened by their inclusion (in Supplementary materials). That said, some methodological assertions were a little cryptic; on line 517, I was unclear how Figure S3 speaks to the topographic distribution of the signal? I was heartened to see a reference an online code repository, but I would encourage the authors to make that publicly accessible, and therefore easily reproduced. Ultimately, given the (relative) simplicity of the use of the EEG data as the comparison (and raw input) data for the modelling efforts, these methodological details are of secondary concern.

I am less able to speak to the details of the modelling approaches used, which ultimately lie at the heart of the contribution. At the broad level of description afforded in the Supplementary section, the approach described sounded sensible, cautious and well justified. However, my expertise is insufficient

to speak to the instantiation here. All elements in the modelling section are generally well characterized, refreshingly directly explained and appear reproducible.

We would like to thank both Reviewers for their insightful suggestions and comments. We found their reviews to be highly constructive and indicative of their expertise in the area. Thanks to the reviewers, the revised manuscript is a significant improvement on the original.

Verbatim reviewer's comments are in **blue**, our responses to the comments are in **black**, and the corresponding amendments to the manuscript are in **green**.

Reviewer #1

This paper explores the coding of orientation in the brain using a forward model (similar to Brouwer/Heeger) and various the population parameters to fit the data, with the best-fitting model having a denser collection of tuning curves at the cardinal orientations, with unequal skewing of the density for horizontal vs. vertical. I was fairly amazed that this could work using EEG data (I have no experience with that).

I'm also a bit confused by what the result means in terms of cortical coding, since the collection of decoded electrodes presumably spanned all of early visual cortical regions and beyond. When folks first started decoding orientation with fMRI in V1 (Frank Tong), folks on my floor complained that the reason it worked was not that voxels have slight random biases toward one orientation column or another, but rather that there was an overall regional bias depending on polar angle in the retinotopic map (Freeman/Merriam/Heeger), and that debate continues to this day. You are using EEG signals, which pool over a far broader region per sensor, so sure, you can decode, but why can you do so and why does it make sense to relate those signals to a channels model based on tuning in V1 alone?

We agree that the signals recorded in our study represent the aggregate activity across primary visual cortex and beyond. Similar to the explanation for orientation decoding from fMRI signals, which Reviewer 1 describes, we think that orientation decoding from EEG signals is possible on the basis of biases, e.g., the cardinal bias and the radial bias, that are differentially present across the visual cortex. These biases seem to elicit sufficiently different patterns of activation in response to different orientations across the 20 occipital and parietal sensors to facilitate reliable decoding. Regarding the question of why it makes sense to relate these signals as to the tuning of V1 orientation-tuned neurons, we did this primarily for simplicity. Reviewer 1 is absolutely correct that other neurons, e.g., those in V2 and beyond, also likely contributed to the results and the tuning of these neurons may be more complex; although, Taylor & Bays (2020) suggest that the tuning functions of neurons sensitive to visual orientation don't vary much across the visual cortex. We are unable to confidently distinguish between these sources of activity with EEG, fMRI would be better suited for this. To the extent that neurons with more complex tuning respond to orientation, we think that biases present in V1 are likely inherited by later regions, e.g., V5 (Xu et al, 2006, PNAS), and less likely to have introduced qualitatively different biases.

However, as we cannot rule this possibility out, we have acknowledged this limitation in the Discussion of the revised manuscript.

The neural responses recorded in the current study with EEG represent activity from primary visual cortex in addition to other upstream visual areas (e.g., V2, V3, etc.). Thus, while we modelled the orientation tuning functions of neurons based on the known properties of V1 neurons, it is likely that other neurons within the visual processing cascade, which have more varied tuning properties, also contributed recorded representation. While neurophysiological work suggest similar orientation biases are present in higher visual regions, e.g., V5/MT³⁹, future work using an imaging technique with better spatial specificity, e.g., fMRI, could test whether the same prior is instantiated across different cortical areas.

Regardless, this is an interesting paper and a real contribution.

We appreciate Reviewer 1's recognition of the contribution of the study.

Comments by line number:

81: "from the grand average of all gratings" - I suppose this should be clarified as "from the grand average of responses, averaged over responses to all gratings" or some such. The same applies to line 94.

This has been clarified in the revised manuscript.

Line 82: We first characterized orientation-related univariate activity. We sorted the gratings into six orientation bins ($\pm 15^\circ$ around 0° , 30° , 60° , 90° , 120° , and 150° , where 0° is horizontal) and calculated the difference between the average evoked response for each bin and the grand average of responses, obtained by averaging over responses to all gratings.

Figure 2b caption: b) Difference in event related potentials for gratings within orientation bins ($[0^\circ, 30^\circ, 60^\circ, 90^\circ, 120^\circ, 150^\circ] \pm 15^\circ$) and the grand average of responses, obtained by averaging over responses to all gratings.

117: Looking for bias averaged over orientations seems a bit silly, eh?

Yes, we agree that this analysis would be unlikely to yield any significant biases. We included it here for completeness and transparency (and also as a sanity check) with the in-text caveat that we did not expect to find anything. However, if Reviewer 1 thinks that the manuscript would be significantly improved by the removal of this analysis, then we would be happy to comply.

Figure 3h looks a lot different from 3e and those three peaks are kind of bizarre, yes? Nevertheless, these results from extant data (the replication) are quite impressive.

Yes, the precision results look quite different between datasets. However, we think this is a feature, not a bug, of the analysis. That is, while the task we employed was orthogonal to the orientation content of the stimulus (spatial frequency detection), King & Wyart (2021) instructed participants to count the number of cardinal and oblique orientations that appeared during the sequence. We think that the additional peaks at cardinal and oblique orientations in their data reflect attention to these orientations, caused by task demands. We hope that this incidental finding means that the analysis developed in the current study will also be useful for investigating cognitive processes, such as attention. We also agree that given the differences in temporal dynamics between the design of the two experiments, correspondence between the two datasets is quite striking.

Line 164: Note that the observers' task in the previous study was to report whether grating stimuli were more cardinally or obliquely oriented on a given trial, likely resulting in increased precision around those orientations (Fig. 3h). This re-analysis therefore demonstrates that our neural decoding method is sensitive to task-related goals, with replicable estimates of orientation anisotropies in tuned responses and bias.

210: responses and biases ... were equivalent: That was guaranteed as your model was symmetric in that way.

Yes, we agree that this was guaranteed. But this symmetrical model is the standard representation of the cardinal bias in visual cortex. Although we expected that it would produce symmetrical vertical and horizontal biases, we thought it important to directly test the capacity of the standard models to explain the empirical results. We have acknowledged that this symmetry was entirely expected in the revised manuscript.

Line 215: In contrast to the empirical results of the two datasets, however, the peak responses and biases around horizontal and vertical orientations were equivalent, which was expected from cardinally symmetric response functions, revealing that the modelled response functions did not fully reflect the neural architecture underlying orientation anisotropies in human visual cortex.

351: At this point you haven't yet pointed out that, although the usual Bayesian setup suggests attraction to the cardinals, that behavior in estimation tasks (unlike my discrimination task with Girshick and Simoncelli) often finds repulsion from the cardinals. Your discussion on Wei/Stocker on this seems a bit thin, since you have a setup that guarantees relatively symmetric likelihoods, and thus your model won't produce the repulsion in estimates that Wei/Stocker think is correct

and, in their cost function, optimal. You do talk about this on the next page, and you do get this from your asymmetric model, but I don't know that the literature finds asymmetries with attraction to horizontal and repulsion from vertical or vice versa (although I thought Frank Durgin once told me that was a finding and folks related it to something about ground planes in the environment ... whatever...).

Reconciling the discrepancy between neural data showing attractive biases versus psychophysical reproduction tasks that result in repulsive responses remains an ongoing goal of our team. Importantly, the model we invoke can produce attractive or repulsive responses, depending on the ratio of internal vs external noise in the model – Wei and Stocker (2015) made the same observation (see Fig. 5 in their paper). We are currently collecting a large dataset that specifically tests whether repulsive vs attractive responses are different for targets around horizontal and vertical, and how any such biases are influenced by internal versus external noise. Our preliminary evidence does indeed suggest that there are different biases around vertical vs horizontal, but these experiments are ongoing and beyond the scope of the present manuscript.

429: Hmmm, I'd think that more text should be devoted to this massive drop in participants you analyzed.

The data analyzed in this study was collected alongside of another dataset in which we investigated the tilt aftereffect. As such, we included an initial, purely behavioural, screening session (on a different day) in which participants performed a similar task to the one in the main experiment (spatial frequency detection) in addition to a reproduction task used to measure an induced tilt aftereffect. Despite the robustness of the classic tilt aftereffect, we found that almost half of participants failed to show a significant effect and were not included in the main EEG experiment. Thus, with the exception of one participant who's data was not analyzed, due to hardware failure, all participants who were included in the EEG experiment were included in the analyses. However, we agree that this section was light on detail and have provided additional details in the revised manuscript.

Line 441: Prior to the main experiment, participants were required to pass an initial screening session in which they completed the same task but without EEG. To pass the screening, participants were required to perform above chance (25%) on the detection task. Further, they were required to show a tilt aftereffect (induced by adaptation) on a separate orientation reproduction task (assessed using a one-tailed *t*-test), which was related to data collected for a different study. This resulted in approximately half of the participants being screened out (32 due to insufficient repulsive bias, and 8 due to both insufficient repulsive bias and low detection task performance). Screening and experiment sessions were separated by a minimum of 24 h. Note that the individuals who were screened out are not included in the *Participants* section.

474: I would have liked to see more text here about this analysis method and, in particular, how you get enough data to estimate a covariance matrix (which usually needs more data than one can gather to get a stable estimate).

We followed the procedure developed and validated by (Kok, Mostert & de Lange, 2017). We have included a more detailed description of the analysis method in the revised manuscript.

Line 501: To compute the inverse model, we estimated the weights that, when applied to the data, would reconstruct the underlying channel activities with the least error. In line with previous magnetencephalography work^{24,49}, when computing the inverse model, we deviated from the forward model proposed by²¹ by taking the noise covariance into account to optimize it for EEG data, given the high correlations between neighbouring sensors. We then estimated the weights that, when applied to the data, would reconstruct the underlying channel activities with the least error. Specifically, \mathbf{B} and \mathbf{C} were demeaned such that their average over presentations equalled zero for each sensor and channel, respectively. The inverse model was then estimated using a subset of the data, selected through cross-fold validation (10 folds). The hypothetical responses of each of the five channels were calculated from the training data in each fold, resulting in the response row vector $\mathbf{c}_{train,i}$ of length n_{train} presentations for each channel i . The weights on the sensors \mathbf{w}_i were then obtained through least squares estimation for each channel:

$$\mathbf{w}_i = \mathbf{B}_{train} \mathbf{c}_{train,i}^T (\mathbf{c}_{train,i} \mathbf{c}_{train,i}^T)^{-1} \quad (2)$$

where \mathbf{B}_{train} indicates the (m sensors \times n_{train} presentations) training EEG data. Subsequently, the optimal spatial filter \mathbf{v}_i to recover the activity of the i th channel was obtained as follows⁴⁹:

$$\mathbf{v}_i = \frac{\tilde{\Sigma}_i^{-1} \mathbf{w}_i}{\mathbf{w}_i^T \tilde{\Sigma}_i^{-1} \mathbf{w}_i} \quad (3)$$

where $\tilde{\Sigma}_i$ is the regularized covariance matrix for channel i . Incorporating the noise covariance in the filter estimation leads to the suppression of noise that arises from correlations between sensors. The noise covariance was estimated as follows:

$$\tilde{\Sigma}_i = \frac{1}{n_{train} - 1} \boldsymbol{\varepsilon}_i \boldsymbol{\varepsilon}_i^T \quad (4)$$

$$\boldsymbol{\varepsilon}_i = \mathbf{B}_{train} - \mathbf{w}_i \mathbf{c}_{train,i} \quad (5)$$

where n_{train} is the number of training presentations. For optimal noise suppression, we improved this estimation by means of regularization by shrinkage using the analytically determined optimal shrinkage parameter⁴⁹, yielding the regularized covariance matrix $\tilde{\Sigma}_i$.

487-497: I found this text to be opaque.

We agree that this text was not very clear in the original manuscript. We thank Reviewer 1 for highlighting this to us and have clarified this section of the revised manuscript.

Line 531: The ‘enhanced inverted encoding model’ (eIEM) method involves repeatedly computing the inverse model while rotating the forward model channel orientation preferences until each orientation is represented by a channel response (e.g., 180 channels, one for each orientation). The eIEM method was established using data with feature identities sampled from discrete bins (e.g., 0°, 30°, 60°, 90°, 120° and 150°), which determine: *a*) the number of channels used in each model (e.g., 6), and *b*) the number of feature identities from which predictions could be derived from the dense output (e.g., 6). By contrast, we presented observers with orientations sampled from a continuous uniform distribution of all orientations. Thus, we first sorted presentations into five orientation bins (0°, 36°, 72°, 108°, and 144°). After rotating channels to produce 180 channel responses for each presentation, we then binned the responses according to the original grating orientation (rounded to nearest the nearest degree); that is, the orientation of the presentation prior to binning. This allowed us to produce high-resolution model predictions for each orientation.

Eq. 2: The equation is a bit of a shorthand since \mathbf{c} is a vector and you are like the $\exp()$ is also producing a vector, so that this is a complex dot product, yes? Makes sense, but it's a shorthand that might trip people up. In any case, z is a complex number with a length that truly is a kind of response strength, but it isn't what you use. Instead, you use \hat{r}_θ . This latter value (Eq. 4) isn't a strength per se. Rather, you average (you need to state what trials you are averaging over) unit complex difference vectors and ask what is the component in the "correct" direction and call that a response strength. That seems a misnomer to me. Much of this page is standard circular statistics (adapted to the range 0- π) and could use a little clarification for those who are unfamiliar with this.

We have clarified this section in the revised manuscript by expanding equation 6 and explicitly stating that we measure the strength of the response at the presented orientation (referring to this index as “orientation response”) and how we averaged the data.

Line 557: For each presentation, we decoded orientation by converting the channel responses to polar form:

$$\begin{aligned} z &= \mathbf{c} \cdot e^{2i\varphi} \\ &= \sum_k c_k \exp(2i\varphi_k) \end{aligned} \quad (6)$$

and calculating the estimated angle:

$$\hat{\theta} = \frac{\arg(z)}{2} \quad (7)$$

where \mathbf{c} is a vector of channel responses and $\boldsymbol{\varphi}$ is the vector of angles at which the channels peak (multiplied by two to project 180° orientation space onto the full 360° space). From the decoded orientations, we computed the mean resultant vector of the difference between decoded and grating orientations, averaged across presentations of the same grating orientation:

$$\bar{R} = \frac{1}{n} \sum_{j=1}^n \exp(i(\hat{\theta}_j - \theta))$$

The direction and amplitude of this vector quantity reflected the accuracy and variability of the decoding, respectively.

Based on the resultant vector we calculated three estimates: *orientation tuned response*, *precision*, and *bias*. Orientation tuned responses represented the amplitude of the response at the presented orientation²⁴, and were expressed by projecting the mean resultant onto a vector with 0°:

$$\hat{r}_\theta = \text{Re}[\bar{R}], \quad (8)$$

Precision was estimated by calculating the angular deviation (a circular analogue to standard deviation⁵¹) of the decoded orientations within each orientation bin:

$$\hat{\sigma}_\theta = \sqrt{2(1 - |\bar{R}|)} \quad (9)$$

and normalizing, such that values ranged from 0 to 1, where 0 indicates a uniform distribution of decoded orientations across all orientations (i.e., chance-level decoding) and 1 represents perfect consensus among decoded orientations:

$$\hat{p}_\theta = 1 - \frac{2\hat{\sigma}_\theta}{\sqrt{2}} \quad (10)$$

Bias was estimated by computing the circular mean of angular difference between the decoded and presented orientation:

$$\hat{b}_\theta = \arg(\bar{R}) \quad (11)$$

532: I'm not sure what "the tuning curve of orientation θ " means, but c_θ is, rather, the set of tuning curves evaluated at orientation θ .

We thank Reviewer 1 for noticing this ambiguity in the description. We have amended this in the revised manuscript.

Line 599: c_θ indicates the set of tuning curves evaluated at orientation θ

535-542: This was a bit unclear as well; be kind to your readers ;^)

This was a slightly challenging process to explain in writing and we agree that the original version lacked sufficient clarity to be easily understood. We have clarified this section in the revised manuscript in order to improve interpretation.

Line 604: Uneven *tuning preferences* were modelled by shifting the neural tuning functions according to the sum of two von Mises derivative functions ($\kappa=0.5$) centered on 0° (amplitude=15) and 90° (amplitude=10), such that the maximum shift applied was 15° . This resulted in shifting the neural response functions towards the cardinals, with increased clustering around horizontal relative to vertical. Uneven *tuning widths* were modelled by narrowing the neural tuning functions according to the sum of two von Mises functions ($\kappa=2$) centered on 0° (amplitude=15) and 90° (amplitude=10), such that the maximum increase to κ was 15. This resulted in narrowing the neural response functions that preferred cardinal orientations, with increased narrowing around horizontal relative to vertical.

Reviewer #2

Thank you for the opportunity to review this paper. Wonderful to see EMCRs conducting work of this quality. Its coupling of several technically sophisticated analytical methods allowed the authors to uniquely identify the specific form of orientation anisotropy instantiated in the human visual cortex. Viewed in isolation, the contribution of this article appears narrow, in that it merely resolves the disputed parameterization of orientation biases well-known to exist in human visual processing. But this is the wrong lens with which to view this data, because while it largely answers these outstanding questions, its novel methodology also answers all the intervening questions and thereby presents methods and insights likely useful beyond psychophysics and cognitive neuroscientists. These insights are afforded by viewing and quantifying the data using cumulatively novel modelling approaches. The visual communication of these methods and their findings with clean, neat figures was particularly commendable, as it demonstrates the authors' novel perspective on this issue immediately. The observation that humans have a vertical-biased, cardinal orientation preference which aligns with what is observed in decomposed of natural scenes is an insight useful to those working on inferential statistics, computer vision and evolutionary biology. Our visual system acts as if it possesses the same biases as would be needed in an efficient Bayesian estimator trained to process natural scenes, but not scenes of constructed architecture. This is a fundamentally interesting observation.

We appreciate Reviewer 2's comments on the manuscript. While we think that our findings address an important empirical question in visual neuroscience, given the broad applicability of the methods developed in the study, we also hope that the manuscript will have impact beyond visual neuroscience.

The work generally appears sufficient to support the claims. The management of the EEG data appears sound, although more could have been done to explain the precise transformations and cleaning methods used. What was the ERP baseline?

With the exception of high-pass filtering, no additional baselining was applied to the ERP; however, note that data were demeaned in a timepoint per timepoint manner during

forward encoding analyses. We have included the grand average and sensor-level ERP data in the Supplementary Material of the revised manuscript.

Figure S1. Event-related potentials. a) Grand average event-related potential, averaged across occipital and parietal sensors (cyan dots of inset), trials, and participants for the stimulus epoch (-50 to 500 ms). Event-locked and subsequent gratings indicated by solid and dashed black rectangles, respectively. b) Same as (a), but separately for each sensor (coloured dots of inset indicate sensor locations). Shaded regions in (a) indicate SEM across participants.

What parameters were used in the artifact cleaning toolboxes called?

We have specified the parameters used to perform artifact cleaning and electrical line noise removal in the revised manuscript.

Line 458: Using EEGLAB⁴⁴, electrical line noise was removed using *pop_cleanline.m* with the following parameters: *bandwidth=2; all channels; linefreqs=[50 100 150 200 250]; normSpectrum=0; p=0.01; pad=2; scanforlines=true; sigtype=channels; tau=100; winsize=4; winstep=4*. The *clean_artifacts.m*⁴⁴ function (default parameters) was used to remove bad channels (identified using Artifact Subspace Reconstruction)...

What kind of rejection and interpolation was used (per channel and in ASR)?

We have specified the kind of rejection and interpolation used in the revised manuscript.

Line 462: ...was used to remove bad channels (identified using Artifact Subspace Reconstruction), which were then spherically interpolated from the neighbouring electrodes. Only individual channels were removed and interpolated, no time periods were removed from the data.

In my experience, these parameters can strongly impact on the data patterns that emerge.

We agree that these parameters can influence the pattern of results and that it is important to declare them in the interest of transparency and reproducibility. However, the influence of preprocessing choices seems to be more significant for waveform analyses than decoding analyses, which typically treat data at different timepoints independently and include additional normalization steps.

The choice of analysis window, while data-driven, was slightly unexpected. The quantification window (50-400ms) runs into the presentation of the subsequent stimulus, and so too does the baseline window (-250ms) ...

One interesting finding to come from the decoding literature, coincidentally from the study that includes the data we used to replicate our findings (King & Wyart, 2019), is that neural representations of visual features (e.g., orientation) seem to be quite robust to additional incoming information. Indeed, as King & Wyart (2019) show, and we see in our own data, we can reliably decode the orientation of gratings even after two subsequent different gratings have been presented. We have now commented on this in the revised manuscript.

Line 114: Orientation tuned responses rose sharply from ~50 ms following stimulus onset and gradually reduced over the following 400 ms (Fig. 3a, cyan data), revealing that decodable information is relatively stable across time and robust to additional incoming information, i.e., subsequently presented stimuli, consistent with recent work^{23,24}.

, although I did not see any evidence that baselining took place (Fig 2b also implies some variation in ERP prior to stimulus onset).

With the exception of high-pass filtering, no additional baselining took place. We hope that the inclusion of the raw ERPs in the supplementary material speak to this point.

Given the caution evident in the authors' choice of time window and electrode set (Figures S1 and S2 were welcome additions), which exceeds standards in the field, I am confident there are principled reasons for this design choice but the article would be strengthened by their inclusion (in Supplementary materials). That said, some methodological assertions were a little cryptic; on line 517, I was unclear how Figure S3 speaks to the topographic distribution of the signal?

This was a typo, we thank Reviewer 2 for spotting it. It was meant to be a reference to Figure S2 (formerly Figure S1), which shows the topographic response to orientations. This has been amended in the revised manuscript.

I was heartened to see a reference an online code repository, but I would encourage the authors to make that publicly accessible, and therefore easily reproduced. Ultimately, given the (relative)

simplicity of the use of the EEG data as the comparison (and raw input) data for the modelling efforts, these methodological details are of secondary concern.

The online code repository has now been made public.

Line 710: Code Availability. Code has been deposited in GitHub (<https://github.com/ReubenRideaux/Neural-tuning-instantiates-prior-expectations-in-the-human-visual-system>).

I am less able to speak to the details of the modelling approaches used, which ultimately lie at the heart of the contribution. At the broad level of description afforded in the Supplementary section, the approach described sounded sensible, cautious and well justified. However, my expertise is insufficient to speak to the instantiation here. All elements in the modelling section are generally well characterized, refreshingly directly explained and appear reproducible.

We thank Reviewer 2 for these comments. Given the novelty of the methods and our desire to have them applied by other research groups, we endeavoured to describe the methods so as to facilitate reproduction.

REVIEWERS' COMMENTS

Reviewer #1 (Remarks to the Author):

Re-review of "Neural tuning instantiates prior expectations in the human visual system"

I was impressed by the paper last time and still am. I'm still amazed that global orientation biases picked up by EEG are enough to get this much mileage in thinking about underlying biased tuning. The paper has been clarified as requested and my comments at this point are just trivia. Well done!

28: Those references in the parenthesis should be reference numbers

Fig. 5 legend: in indicate -> indicate

498: five channels -> six channels? In Methods there is some discussion of switching to five channels that I didn't quite parse, but here it comes out of the blue and seems like a typo

538: "represented the strength of the response": Not really, the strengths of the channel responses are dropped as soon as you compute $\arg(z)$. The orientation tuned response is more about how close the responses are typically to the correct answer (average absolute bias), not about the strength of the channel responses as decoded per se.

572: "assigned from a uniform distribution": This should yield information about the underlying channels that is far greater than the mixing I'd expect from huge numbers of neurons picked up by EEG electrodes. As you say later (and as I prompted in the previous review), the signal about the channels you get on EEG isn't about random alignment of tuning preferences to electrodes, but rather about gross biases in the tuning of the population over large swathes of V1/V2/V3 (as was discussed in the Merriam/Freeman/Heeger papers about why decoding from fMRI works).

632: 14 -> 18

640: 16 -> 20

Reviewer #2 (Remarks to the Author):

All comments were directly and thoughtfully addressed. I have no remaining concerns. I look forward to reading the final article. Well done.

Verbatim reviewer's comments are in blue, our responses to the comments are in black, and the corresponding amendments to the manuscript are in green.

Reviewer #1

I was impressed by the paper last time and still am. I'm still amazed that global orientation biases picked up by EEG are enough to get this much mileage in thinking about underlying biased tuning. The paper has been clarified as requested and my comments at this point are just trivia. Well done!

We appreciate the Reviewer 1's continued commendations of the paper.

28: Those references in the parenthesis should be reference numbers

Fixed.

Fig. 5 legend: in indicate -> indicate

Fixed.

498: five channels -> six channels? In Methods there is some discussion of switching to five channels that I didn't quite parse, but here it comes out of the blue and seems like a typo

This was a typo and has been corrected to state "six channels" in the revised manuscript.

538: "represented the strength of the response": Not really, the strengths of the channel responses are dropped as soon as you compute $\arg(z)$. The orientation tuned response is more about how close the responses are typically to the correct answer (average absolute bias), not about the strength of the channel responses as decoded per se.

True – we have replaced the terminology "orientation tuned response" with "accuracy" throughout the manuscript and updated the description in the methods.

Line 451: From the decoded orientations, we computed three estimates: *accuracy*, *precision*, and *bias*. Accuracy represented the similarity of the decoded orientation to the presented orientation²⁴, and was expressed by projecting the mean resultant (averaged across presentations of the same grating orientation) of the difference between decoded and grating orientations onto a vector with 0°

572: "assigned from a uniform distribution": This should yield information about the underlying channels that is far greater than the mixing I'd expect from huge numbers of neurons picked up

by EEG electrodes. As you say later (and as I prompted in the previous review), the signal about the channels you get on EEG isn't about random alignment of tuning preferences to electrodes, but rather about gross biases in the tuning of the population over large swathes of V1/V2/V3 (as was discussed in the Merriam/Freeman/Heeger papers about why decoding from fMRI works).

We agree that the neuron-to-sensor weights used in the generative modelling likely produced more independent information across artificial sensors than that present across the EEG sensors, which are highly correlated. However, we could only reasonably expect this to increase the overall SNR and not have any relationship with orientation. Indeed, we ran simulations in which the variance between neuron-to-sensor weights was reduced and this produced no qualitative difference in the outcome. Thus, we selected the range of 0-1 for simplicity. We have acknowledged this point in the revised manuscript.

Line 486: Neuron-to-sensor weights (w_i) were randomly assigned from a uniform distribution between 0 and 1. This range was selected for simplicity. The variance in weights between the activity of subpopulations of neurons and EEG sensors in the brain is likely different, however, our simulations confirmed that modifying this range had no qualitative influence on the pattern of results.

632: 14 -> 18

Fixed.

640: 16 -> 20

Fixed.

Reviewer #2

All comments were directly and thoughtfully addressed. I have no remaining concerns. I look forward to reading the final article. Well done.

We thank Reviewer 2 for their comments.